# Kinetic Model and Experiment for Self-Ignition of Triethylaluminum and Triethylborane Droplets in Air

**DOI:** 10.3390/mi13112033

**Published:** 2022-11-21

**Authors:** Sergey M. Frolov, Valentin Y. Basevich, Andrey A. Belyaev, Igor O. Shamshin, Viktor S. Aksenov, Fedor S. Frolov, Pavel A. Storozhenko, Shirin L. Guseinov

**Affiliations:** 1Department of Combustion and Explosion, Semenov Federal Research Center for Chemical Physics of the Russian Academy of Sciences, 119991 Moscow, Russia; 2Institute of Laser and Plasma Technologies, National Research Nuclear University “Moscow Engineering Physics Institute”, 115409 Moscow, Russia; 3State Research Center “State Scientific Research Institute of Chemistry and Technology of Organo-Element Compounds”, 105118 Moscow, Russia

**Keywords:** triethylaluminum, triethylborane, oxygen intrusion reaction, rate constant, activation energy, droplet, self-ignition delay, formation of radicals, detailed kinetics, computational code

## Abstract

Triethylaluminum Al(C_2_H_5_)_3_, TEA, and triethylborane, B(C_2_H_5_)_3_, TEB, are transparent, colorless, pyrophoric liquids with boiling points of approximately 190 °C and 95 °C, respectively. Upon contact with ambient air, TEA, TEB, as well as their mixtures and solutions, in hydrocarbon solvents, ignite. They can also violently react with water. TEA and TEB can be used as hypergolic rocket propellants and incendiary compositions. In this manuscript, a novel scheme of the heterogeneous interaction of gaseous oxygen with liquid TEA/TEB microdroplets accompanied by the release of light hydrocarbon radicals into the gas phase is used for calculating the self-ignition of a spatially homogeneous mixture of fuel microdroplets in ambient air at normal pressure and temperature (NPT) conditions. In the primary initiation step, TEA and TEB react with oxygen, producing an ethyl radical, which can initiate an autoxidation chain. The ignition delay is shown to decrease with the decrease in the droplet size. Preliminary experiments on the self-ignition of pulsed and continuous TEA–TEB sprays in ambient air at NPT conditions are used for estimating the Arrhenius parameters of the rate-limiting reaction. Experiments confirm that the self-ignition delay of TEA–TEB sprays decreases with the injection pressure and provide the data for estimating the activation energy of the rate-limiting reaction, which appears to be close to 2 kcal/mol.

## 1. Introduction

Triethylaluminum Al(C_2_H_5_)_3_ (TEA) and triethylborane B(C_2_H_5_)_3_ (TEB) are transparent, colorless, pyrophoric liquids with boiling points of approximately 190 °C and 95 °C, freezing points of approximately −46 °C and −93 °C, and densities of 0.832 and 0.677 g/cm^3^ (at 25 °C), respectively [1,2]. Their solutions remain stable when stored away from heat sources in a dry, inert atmosphere, but, at elevated temperatures, they slowly decompose to form hydrogen, ethylene, and elemental aluminum and boron. Upon contact with air, TEA and TEB and their solutions in hydrocarbon solvents ignite. They also react violently with heated water [3,4,5]. TEA, TEB, and their solutions should only be handled under a dry, inert atmosphere such as nitrogen or argon. TEA is used as a component of the Ziegler–Natta catalyst for the polymerization of olefins [6,7,8]. It is also used in reactions with ethylene for the growth of hydrocarbon radicals at the aluminum atom and, with the subsequent hydrolysis of the resulting higher aluminum alkyls, to obtain fatty a-alcohols. In addition, TEA is used as an alkylating agent in the synthesis of other organoelement and organic compounds. TEB is used in organic chemistry as an initiator in low-temperature radical reactions [9], in the deoxygenation of alcohols [10], and in other processes. Both TEA and TEB are used as a hypergolic rocket propellant, in napalm and incendiary compositions [1,11,12], as well as in micropropulsion [13] and microrobotics [14]. The SpaceX Falcon 9 rocket is known to use a TEA–TEB mixture as a first- and second-stage hypergolic ignitor [15]. TEA–TEB mixtures were also used for motor ignition in the Atlas and Delta commercial launch vehicles. According to [16], “Triethylaluminum market valued at 225.1 million USD in 2020 is expected to reach 255.2 million USD by the end of 2026, growing at a Compound Annual Growth Rate of 1.8% during 2021–2026”.

In experiments with liquid TEA sprayed through a nozzle in air in the form of microdroplets, spontaneous combustion of the mixture is observed. Complete combustion of TEA in air corresponds to the overall reaction [17]
(1)$$2\mathrm{Al}(C_{2}H_{5}{)}_{3}+21O_{2}\;\to \;{\mathrm{Al}}_{2}O_{3}+12{\mathrm{CO}}_{2}+15H_{2}O+Q_{A}$$
where $Q_{A}$ is the heat of reaction (1). The value of $Q_{A}$ is approximately 2444 kcal [2], 2293 kcal [18], and 1955 kcal [19]. The latter value is based on the quantum-mechanical calculation of the structures and energy characteristics of all molecular complexes involved in the TEA self-ignition process. Complete combustion of TEB in air corresponds to the overall reaction
(2)$$2B(C_{2}H_{5}{)}_{3}+21O_{2}\;\to \;B_{2}O_{3}+12{\mathrm{CO}}_{2}+15H_{2}O+Q_{B}$$
where $Q_{B}$ is the heat of reaction (2). The value of $Q_{B}$ is approximately 2100 kcal [20], i.e., it is close to the value of $Q_{A}$. In a complex chemical process of transformation of the initial components into the products of overall reactions (1) and (2), in which the reacting components participate in many heterogeneous and gas-phase elementary reactions, two stages can be distinguished: the stage of self-ignition and the stage of rapid explosive combustion [21]. In applied terms, the kinetic analysis of the former stage seems to be the most important, since it is this stage that determines the time of the entire process. According to the literature, the self-ignition of TEA and TEB in air occurs through radical reactions [17,22,23]. In the primary initiation step, TEA and TEB react with oxygen, producing an active ethyl radical, which can initiate an autoxidation chain in competition with termination or other pathways.

This work deals with the development of a theoretical model and preliminary experimental studies of the first stage, i.e., the self-ignition. Based on the model, the self-ignition delays of TEA–TEB droplets in air at normal pressure and temperature (NPT) conditions are calculated, whereas experiments are used for estimating the Arrhenius parameters of the rate-limiting reaction. For the sake of definiteness, the kinetic analysis is performed for TEA. However, the model proposed herein can be directly applied to the self-ignition of TEB and TEA–TEB mixtures.

## 2. Materials and Methods

### 2.1. Kinetic Model of TEA Droplet Self-Ignition in Air at NPT Conditions

For a kinetic analysis of the self-ignition stage, it is necessary to draw up a scheme of elementary reactions. By definition, the primary reaction in the scheme should be a heterogeneous reaction that occurs when oxygen molecules available in air collide with TEA droplets. Such a reaction, leading to the self-ignition of a mixture occupying a limited volume, should be characterized by a sufficiently low activation energy. Presumably, this may be a heterogeneous reaction of the intrusion of an O_2_ molecule to TEA with the formation of the (C_2_H_5_)_2_Al–O–O–(C_2_H_5_) molecule directly in the collision of TEA and O_2_ molecules
(3)$$\mathrm{Al}(C_{2}H_{5}{)}_{3}+O_{2}=(C_{2}H_{5}{)}_{2}\mathrm{Al}–O–O–(C_{2}H_{5})+Q_{I}$$
or by forming an intermediate complex Al(C_2_H_5_)_3_O_2_ according to the scheme [17]
(4)$$\;\mathrm{Al}(C_{2}H_{5}{)}_{3}+O_{2}=\mathrm{Al}(C_{2}H_{5}{)}_{3}O_{2}\;\to \;(C_{2}H_{5}{)}_{2}\mathrm{Al}–O–O–(C_{2}H_{5})+Q_{I}$$
where $Q_{I}$ = 113 kcal/mol. Reaction (3) or (4) can be followed by the monomolecular decomposition reaction through two channels:(5)$$\;(C_{2}H_{5}{)}_{2}\mathrm{Al}–O–O–(C_{2}H_{5})\;\to \;(C_{2}H_{5}{)}_{2}\mathrm{Al}–O+O–(C_{2}H_{5})+Q_{\mathrm{II}};$$
(6)$$\;(C_{2}H_{5}{)}_{2}\mathrm{Al}–O–O–(C_{2}H_{5})\;\to \;(C_{2}H_{5}{)}_{2}\mathrm{Al}–O–O+C_{2}H_{5}+Q_{\mathrm{III}}$$
where $Q_{\mathrm{II}}$ = −77.9 kcal/mol and $Q_{\mathrm{III}}$ = −90.5 kcal/mol [17]. Reactions (3 or 4) + (5) and (3 or 4) + (6) can be considered as exothermic bimolecular reactions. These reactions occur during collisions of gas-phase molecules with the surfaces of TEA droplets (heterogeneous reactions). In this case, volatile active radicals C_2_H_5_ and C_2_H_5_O enter the gas phase (air) and interact with oxygen, releasing heat and giving rise to other sequential and parallel reactions, the same as in the gas-phase kinetics of the oxidation, self-ignition, and combustion of light alkanes (methane, ethane, and butane) and their derivatives. Kinetic schemes, corresponding equations of chemical kinetics, algorithms, and codes that describe similar gas-phase processes exist, and they can be readily used as subroutines for the numerical solution of the problem under consideration. It is worth emphasizing that we consider only the initial stage of the self-ignition process, rather than the entire process of TEA droplet combustion. At this stage, the size and chemical composition of droplets, as well as the oxygen concentration in the gas, change only a little, and can be considered constant.

Based on this prerequisite, the following kinetic model of TEA self-ignition in air at NPT conditions is proposed. The rate constant of reactions (3 or 4) + (5) and (3 or 4) + (6) is approximated as
(7)$$K=A\exp\left(-\frac{\varepsilon }{RT}\right)$$
where A is the preexponential factor; T is the temperature; and ε is the activation energy. The consumption rate of TEA molecules per unit volume of the mixture can be expressed by the formula
(8)$$\frac{dn_{\mathrm{TEA}}}{dt}=-\frac{n_{O_{2}}u_{O_{2}}}{4}SNw,\;w\approx \lambda \exp\left(-\frac{\varepsilon }{RT}\right)$$
where $n_{\mathrm{TEA}}$ is the number of TEA molecules per unit volume; $n_{O_{2}}$ is the number of oxygen molecules per unit volume; $u_{O_{2}}$ is the thermal velocity of oxygen molecules; $\frac{n_{O_{2}}u_{O_{2}}}{4}$ is the number of collisions of oxygen molecules with a unit surface of a droplet per unit time; $S=4\pi r_{d}^{2}$ is the surface area of a TEA droplet; $r_{d}$ is the TEA droplet radius; N is the number of TEA droplets per unit volume; w is the reaction probability in one collision; and λ is the steric factor. This latter factor is unknown. Its value is probably in the range of 0.1–0.01. According to the kinetic theory of gases, the thermal velocity $u_{O_{2}}$ at temperature T is
(9)$$u_{O_{2}}={\left(8RT/32\pi \right)}^{1/2}$$

By definition, the derivative $\frac{dn_{\mathrm{TEA}}}{dt}$ can be also expressed as
(10)$$\frac{dn_{\mathrm{TEA}}}{dt}=-Kn_{O_{2}}n_{\mathrm{TEA},s}=-KSd_{1}n_{O_{2}}n_{\mathrm{TEA},d}N$$
where $d_{1}$ is the thickness of the outer molecular monolayer in a TEA droplet; $Sd_{1}$ is the volume of the outer molecular monolayer in a TEA droplet; $n_{\mathrm{TEA},d}$ is the number of TEA molecules per unit droplet volume; $n_{\mathrm{NEA},s}$ is the number of TEA molecules in the volume $Sd_{1}N$. Substituting Equation (7) into Equation (8) and comparing Equation (8) with Equation (10), one obtains
(11)$$A=\frac{\lambda u_{O_{2}}}{4n_{\mathrm{TEA},d}d_{1}}$$

Equation (11) has a simple physical meaning. The values of $d_{1}$ and $n_{\mathrm{TEA},d}$ are expressed in terms of the effective radius, $r_{1}$, of the TEA molecule: $d_{1}=2r_{1}$, $\frac{1}{n_{\mathrm{TEA},d}}=4\pi r_{1}^{3}/3$. From here and from Equation (11), one obtains
(12)$$A=\left(\frac{2}{3}\right)\lambda u_{O_{2}}\pi r_{1}^{2}=\left(\frac{2}{3}\right)\lambda u_{O_{2}}\sigma$$
where $\sigma =\pi r_{1}^{2}$ is the effective collision cross-section. Formula (12) coincides with the definition of the preexponential factor in the thermal theory of the rate constants of bimolecular reactions in gases [24] up to a factor of 2/3. It follows from Equations (7), (10), and (11) that
(13)$$\frac{dn_{\mathrm{TEA}}}{dt}=-\frac{\lambda }{4}u_{O_{2}}SNn_{O_{2}}\exp\left(-\frac{\varepsilon }{RT}\right)$$

It follows from Equation (13) that the rate of reactions (3 or 4) + (5) and (3 or 4) + (6) depends on the TEA droplet size, $r_{d}$, the oxygen concentration in the environment, $n_{O_{2}}$, and the local instantaneous air temperature, T:(14)$$\frac{dn_{\mathrm{TEA}}}{dt}\sim \frac{n_{O_{2}}}{r_{d}}\exp\left(-\frac{\varepsilon }{RT}\right)$$

Therefore, it could be expected that the self-ignition delay of TEA spray in ambient air could be a function of the spray injection pressure, as the droplet diameter generally depends on the injection pressure: the higher the injection pressure, the smaller the droplet size and the shorter the self-ignition delay. According to Equation (14), the self-ignition delay is shorter if the environment contains more oxygen, and if the local instantaneous air temperature is higher.

In addition to the uncertainty in the value of λ, the rate constant (7) contains an unknown activation energy ε. There exist empirical formulae establishing the relationship between ε and the heat of the exothermic reaction, Q, in a linear approximation:(15)ε=a−bQ
with positive parameters a and b. The values of a and b vary depending on the type (set) of reactions. These formulae include the well-known Polanyi–Semenov rule [24]:(16)ε=11.5−0.25Q kcal/mol 

This rule, when applied to bimolecular reactions (3 or 4) + (5) and (3 or 4) + (6), gives, respectively,
(17)ε=2.7 and 5.9 kcal/mol

As noted in [21], the formulae such as (16) must be used with great caution. The same is true for estimates (17). They can only be considered as a rough approximation, which must be verified and refined experimentally. In experiments, the parameters of the reaction rate in Equation (13) are not measured directly. However, the induction period before the self-ignition of TEA droplets and some other kinetic and thermodynamic parameters can be measured, which depend on the rate constant (7). In this case, the activation energy ε can be found by solving the inverse problem. To do this, one must first solve the direct problem, which consists in calculating the self-ignition delay with a variation in activation energy ε.

### 2.2. Self-Ignition Delays for C_2_H_5_–Air and C_2_H_5_O–Air Mixtures

The mathematical statement of the problem is the statement of the standard problem of the self-ignition of a gas mixture [25] with a given detailed kinetics [26], which is supplemented by a heterogeneous mechanism for the formation of C_2_H_5_ or C_2_H_5_O radicals. The equations for the conservation of the energy and mass of the components have the form
(18)$$c_{p}\rho \frac{dT}{dt}=\Phi$$
(19)$$\rho \frac{dY_{j}}{dt}=w_{j}+\Psi ,\;j=1,2,\ldots ,M$$
where t is time; M is the number of components in the gas mixture; $Y_{j}$ is the mass fraction of the jth component; $c_{p}$ is the heat capacity of the gas mixture at constant pressure; ρ is the density of the mixture; Φ is the heat release in chemical reactions; $w_{j}$ is the component consumption in chemical reactions; and Ψ is the formation of C_2_H_5_ or C_2_H_5_O in reaction (6) or (5), respectively. The system of Equations (18) and (19) is supplemented by the ideal gas equation of state, expressions for Φ and $w_{j}$ [27], and by the polynomial relationship for the heat capacity, while the polynomial coefficients are taken from [28].

Compared to the standard problem formulation for gas mixture self-ignition, the expression for Ψ and all other considerations associated with this circumstance are new. It is assumed that the mixture is initially represented by pure air, and radicals C_2_H_5_ or C_2_H_5_O appear in the gas due to the heterogeneous reaction (6) or (5), respectively. Self-ignition delays depend on the rates of formation of C_2_H_5_ and C_2_H_5_O radicals in heterogeneous reactions (6) and (5) and on the rates of their interaction with oxygen in the gas phase. As both reactions, (6) and (5), are possible, for determining the effect of these radicals on the self-ignition delay, the problem must be solved for two options: (*i*) for a C_2_H_5_–air mixture and (*ii*) for a C_2_H_5_O–air mixture.

#### 2.2.1. Option *i*: C_2_H_5_–Air Mixture

The expression for Ψ is
(20)$$\Psi =\{\begin{array}{c} W_{C_{2}H_{5}}\frac{dn_{C_{2}H_{5}}}{dt}{\;\mathrm{for}\;C}_{2}H_{5}\\ 0\;\mathrm{for}\;\mathrm{other}\;\mathrm{species} \end{array}\;$$

Here, $W_{C_{2}H_{5}}$ is the molecular mass of C_2_H_5_; $\frac{dn_{C_{2}H_{5}}}{dt}$ is the rate of change in the concentration (mol/cm^3^/s) of C_2_H_5_ radicals in the heterogeneous reaction (6), which, in accordance with reaction (6) and Equation (13), satisfies the equation
(21)$$\frac{dn_{C_{2}H_{5}}}{dt}=\frac{\lambda }{4}u_{O_{2}}SNn_{O_{2}}\exp\left(-\frac{\varepsilon }{RT}\right)$$

#### 2.2.2. Option *ii*: C_2_H_5_O–Air Mixture

For this option, the same system of equations is solved as for option *i*, but in Equations (20) and (21) everywhere in the rows and in the indices, C_2_H_5_ must be replaced by C_2_H_5_O. To solve the problem for these two options, a special computational code has been developed.

### 2.3. Experimental Setup

Before designing the experimental setup, it was implied that the self-ignition delay, $\tau_{i}$, of the TEA/TEB spray in ambient air could be estimated experimentally either by measuring the time delay between the start of spray injection and the appearance of self-ignition luminosity, if TEA/TEB is sprayed in a short pulse mode, or by measuring the width of the dark zone between the injector nozzle face and the luminous combustion plume, if the TEA/TEB is sprayed continuously. It could be assumed that ignition occurs much later than the pulsed injection of TEA/TEB. During this time, TEA/TEB droplets slow down, causing the ignition to occur in a virtually quiescent and spatially homogeneous mixture. With an average droplet path of around 10 cm and a speed of escape from the injector nozzle of the order of 10^4^ cm/s, the deceleration time is around 1 ms. This time is much shorter than the expected self-ignition delay time, even at the lowest estimated value of activation energy in Equation (17), ε≈2 kcal/mol. With the continuous spraying of TEA/TEB into the ambient air, one could expect the appearance of a quasi-stationary luminous combustion plume. In this case, TEA/TEB droplets ignite due to the air flow around them, and the self-ignition delay, $\tau_{i}$, of the spray is determined by the time given to a droplet to enter the zone of the luminous combustion plume. The value of $\tau_{i}$ can be estimated experimentally based on the width, L, of the dark zone between the injector nozzle face and luminous combustion plume measured along the spray axis. The value $\tau_{i}$ is related to L and the speed of the TEA/TEB spray at the nozzle exit, U, as $\tau_{i}\approx L/U$.

Based on these implications, the experimental setup for fuel spraying in air was designed and manufactured. The experimental setup consisted of an electromagnetic fuel injector (BOSCH 0 280 158 017) and a system for ensuring its operation (hydraulic system and microprocessor control unit), an optical system, a high-speed camera (Phantom Miro LC310), and a safety system. The elements of the setup are shown in Figure 1a. The injector nozzle had 4 holes with a diameter of 0.2 mm. The microprocessor control unit monitored the current through the injector and the voltage applied, and issued synchronization and trigger signals for the high-speed video camera. In the preliminary experiments, the standard 13%TEA–87%TEB mixture provided by the production company was used. The density of the TEA–TEB mixture was 0.703 g/cm^3^. To prevent clogging of the setup communications by the condensed reaction products of the TEA–TEB mixture with air, the injector was sprayed with n-heptane before and after each experiment. n-heptane (density 0.684 g/cm^3^) was also used for estimating the flow rate and characteristic droplet size in the spray at different injection pressures. The nominal flow rate of n-heptane at an overpressure of 3 and 6 atm was 2.55 ± 0.08 mL/s (1.74 ± 0.05 g/s) and 5.1 ± 0.2 mL/s (3.5 ± 0.1 g/s), respectively. The droplet diameter in n-heptane sprays at the injection overpressure of 3 and 6 atm measured by the slide sampling method [29,30] was ~80–120 and ~30–50 μm, respectively. The operation frequency of the injector in the pulsed mode as well as the injection duration time were varied (see below).

Figure 1b shows the hydraulic scheme of the experimental setup. The experimental procedure was as follows. (i) Before supplying the TEA–TEB mixture to the injector, all communications were thoroughly purged with argon; (ii) liquid n-heptane was poured into the transparent measuring tank through the funnel and pressurized by argon to an overpressure of 3 atm using valve 1; (iii) with valves 2, 4, and 5 closed, valve 3 open, and the injector turned on, the communications were spilled with n-heptane; (iv) valve 3 was closed and the pressure was relieved by briefly turning on the injector; (v) valves 6 and 7 were opened and the TEA–TEB tank was pressurized by argon to the overpressure of 3 to 6 atm; valve 8 was used to control the pressure level; (vi) valves 6 to 8 were closed and valves 1, 4, and 5 opened; (vii) using a rotameter, a small flow rate of argon (around 1 L/min) was established around the injector nozzle to avoid nozzle clogging by the condensed reaction products of the TEA–TEB mixture with air available in the vicinity of the nozzle face; and (viii) the injector was turned on and operated for a preset time either in the pulsed or continuous injection mode.

Figure 2 shows two options used for fastening the injector in the housing, which differed in the means of supplying argon and the geometry of the insulating cavity. The first series of experiments was performed using the recessed injector of Figure 2a and direct video registration of spray self-luminosity during ignition and combustion in ambient air (see Section 3.3). As the dark zone between the flame and the nozzle mouth could be quite short, the second series of experiments was performed using the flat Injector of Figure 2b (see Section 3.3). In the latter case, spray self-ignition and combustion was registered both by direct video registration of self-luminosity and by the schlieren method.

## 3. Results and Discussion

### 3.1. Self-Ignition of Stoichiometric C_2_H_5_–Air and C_2_H_5_O–Air Mixtures at NPT Conditions

Before conducting the calculations for options *i* and *ii*, we calculated the self-ignition delays for stoichiometric C_2_H_5_–air and C_2_H_5_O–air mixtures in the absence of heterogeneous reactions, i.e., at $\Psi_{C_{2}H_{5}}$ = $\Psi_{C_{2}H_{5}O}$ = 0. The overall reaction of C_2_H_5_ with oxygen reads
 4C_2_H_5_ + 13O_2_ → 8CO_2_ + 10H_2_O.

The stoichiometric composition of the C_2_H_5_–air mixture is $X_{C_{2}H_{5}}$ = 0.061; $X_{O_{2}}$ = 0.197; and $X_{N_{2}}$ = 0.742. Self-ignition delays were calculated for the C_2_H_5_–air mixture of stoichiometric composition for NPT conditions: $T_{0}$ = 300 K and p = 1 atm.

The self-ignition delays obtained using the developed kinetic code were compared with the results of calculations using the standard kinetic code KINET, developed by M. G. Neigauz at the Institute of Chemical Physics of the Russian Academy of Sciences for homogeneous gas mixtures [31]. In both cases, the same kinetic mechanism was used: the detailed kinetic mechanism of combustion and oxidation of alkanes up to C4 [26]. It should be noted that the KINET code applies somewhat different thermodynamic data than [28]: the polynomial dependence of heat capacity on temperature has fewer terms.

Table 1 shows the values of the self-ignition delays $\tau_{i}$, the temperature of the reaction products $T_{e}$, volume fractions of C_2_H_5_ radical, ${\left(X_{C_{2}H_{5}}\right)}_{e}$, oxygen, ${\left(X_{O_{2}}\right)}_{e}$, and water, ${\left(X_{H_{2}O}\right)}_{e}$, in the reaction products, as well as the maximum value of methane volume fraction, ${\left(X_{{\mathrm{CH}}_{4}}\right)}_{\max}$, obtained by calculations using the two indicated codes. Figure 3a shows the calculated time histories of temperature obtained by the two indicated codes. Both the data in Table 1 and the curves in Figure 3a are in satisfactory agreement with each other. The slight differences in the results seem to be caused by the differences in the thermodynamic data used.

Then, the self-ignition delays of the stoichiometric C_2_H_5_O–air mixture were calculated under NPT conditions: $T_{0}$ = 300 K and p = 1 atm. The overall reaction of C_2_H_5_O with oxygen reads
 4C_2_H_5_O + 11O_2_ → 8CO_2_ + 10H_2_O.

The stoichiometric composition of the C_2_H_5_O–air mixture is $X_{C_{2}H_{5}O}$ = 0.063; $X_{O_{2}}$ = 0.172; and $X_{N_{2}}$ = 0.765. Figure 3b shows the calculated time histories of temperature for the stoichiometric C_2_H_5_O–air mixture. The self-ignition delays obtained with the new code and KINET are 118 and 124 s, respectively, i.e., the results differ by less than 5%. Thus, calculations show that the self-ignition delay of a stoichiometric C_2_H_5_O–air mixture is much longer than that of a stoichiometric C_2_H_5_–air mixture at NPT conditions. This indicates the much greater reactivity of the C_2_H_5_ radical in air compared to the C_2_H_5_O radical. In view of this, the option with the C_2_H_5_O radical can be omitted.

### 3.2. Self-Ignition of TEA Droplets in Air at NPT Conditions

The problem for option *i* (see Section 2.2) was solved numerically for several values of ε, from 2 to 6 kcal/mol (see Equation (17)), to compare the calculated self-ignition delays with the experiment. For the sake of definiteness, the parameters entering Equation (21) were assumed to have the following values: λ = 0.1; $u_{O_{2}}=4.46\cdot 10^{4}$ cm/s; $r_{d}$ = 50 μm; $S=3.14\cdot 10^{-4}$ cm^2^; *N* = 210 cm^−3^; $c_{p}=1.33\cdot 10^{-3}$ J/(cm^3^K) (see Appendix A and Appendix B). Initial NPT conditions are t = 0; *T* = *T*_0_ = 300 K; pressure p = 1 atm; volume fractions of oxygen and nitrogen $X_{O_{2}}$= 0.21 and $X_{N_{2}}$ = 0.79. All other components: $X_{j}$ = 0; $n_{C_{2}H_{5}}$ = 0; $n_{O_{2}}$ = $X_{O_{2}}$/22,400 = 9.37 · 10^−6^ mol/cm^3^.

Calculations show that the self-ignition delay $\tau_{i}$ depends very strongly on ε (Table 2). Figure 4 shows the calculated time histories of temperature within $0\leq t\leq \tau_{i}$ at ε = 5 and 6 kcal/mol. The upper limit of the specified time interval is chosen equal to $\tau_{i}$ because, at $t>\tau_{i}$, the reactions leading to the complete burnout of TEA, which are not included in the reaction scheme, become significant. Figure 5 shows the calculated time history of the C_2_H_5_ radical concentration, $n_{C_{2}H_{5}}\left(t\right)$, at ε = 6 kcal/mol.

### 3.3. Experimental Results

Figure 6 and Figure 7 show selected video frames of TEA–TEB spray self-ignition during pulsed (Figure 6) and continuous (Figure 7) spraying from the recessed injector of Figure 2a at injection overpressure 3 atm. The duration of the single-shot pulsed spray in the experiment of Figure 6 is 20 ms. After the termination of spray injection in Figure 6, one can see the successive appearance of haze and smoke in the spray core, followed by the formation of a luminous self-ignition spot of a green color, characteristic of TEB combustion (see frame #35 in Figure 6). The first hot spot is located at a distance of around 130 mm from the nozzle face. Subsequently, while this hot spot rapidly grows with time, several other hot spots appear, grow, and overlap with each other, moving in lateral, downstream, and upstream directions and forming a luminous combustion plume. The evolution of the continuous spray in Figure 7 has much in common with that of Figure 6, but the first hot spot is located somewhat closer to the nozzle face (at around 100 mm) and the luminous combustion plume looks more elaborated. The characteristic spray velocity at the nozzle exit estimated based on the mass flow rate and injector nozzle diameter is approximately 20 m/s.

The measured self-ignition delays in Figure 6 and Figure 7 are approximately 30 and 20 ms, respectively. The distance between the nozzle face and luminous plume in a continuous spray appeared to be variable with time (Figure 8) rather than quasi-stationary, as was expected. Thus, while the first ignition event in Figure 8 appeared at a distance of ~100 mm from the nozzle face, it dropped to zero in 100 ms after the start of injection (see Figure 8). This means that the apparent velocity of self-ignition spreading toward the nozzle face was higher than the spray velocity (~20 m/s).

Consider Figure 9 for a better understanding of the circumstances of TEA–TEB spray self-ignition. This figure shows a sequence of synchronized schlieren (left column) and self-luminosity (right column) video frames of TEA–TEB spray injection by the flat injector of Figure 2b, followed by spray self-ignition and combustion. The injection overpressure is 6 atm, and the spray velocity is ~40 m/s. The schlieren images in the left column are superimposed by the self-luminosity images reflected from the mirror of the schlieren system. Schlieren images contain thin droplet trajectories. After a certain delay, the trajectories inflate, forming cloud-like swellings, which can be attributed to the liquid mist and gas, self-heated due to the spontaneous reaction in air. The latter is substantiated by the fact that the luminous self-ignition plume is located exactly in these swellings of the droplet trajectories. Furthermore, Figure 10 shows the pulsed injection of the TEA–TEB spray into the ambient air at NPT conditions with a very short pulse duration of 0.7 ms. Similar to Figure 9, the schlieren images here are also superimposed by the self-luminosity images reflected from the mirror. One can see the appearance of the cloud-like trajectory swellings lagging behind the moving droplets and growing with time in the form of conical tongues of hot matter. It is seen from the schlieren images that the luminous self-ignition plume is also located exactly in these swellings.

Table 3 and Table 4 show the statistics of self-ignition events in the case of four successive injection pulses with a pulse duration of 10 ms and time interval between pulses of 1000 ms (Table 3) and six successive pulses with a pulse duration of 5 ms and time interval between pulses of 400 ms (Table 4). The mean self-ignition delay is seen to be shorter for the conditions of Table 4, thus indicating that the shorter interval between pulses (400 ms vs. 1000 ms) promotes self-ignition, presumably due to the availability of hot residual air on the spray path, which is in line with Equation (14). The results of Figure 6, Figure 7 and Figure 8, as well as Table 3 and Table 4, indicate that the activation energy of TEA–TEB mixture self-ignition in Equation (7) is approximately ε≈2 kcal/mol (see Table 2).

Table 5 and Table 6 show the statistics of self-ignition events in the case of 15 successive injection pulses with a pulse duration of 1 and 2 ms, respectively, and a time interval between pulses of 100 ms. Contrary to Table 3 and Table 4, the injection overpressure for the test fires of Table 5 and Table 6 was 6 atm rather than 3 atm. The mean self-ignition delay is seen to be around 6 ms, which is shorter than that for the lower injection overpressure by a factor of 3 to 5. This result is also in line with Equation (14) as the characteristic droplet size decreases with the injection pressure.

## 4. Discussion

The proposed kinetic model of the self-ignition of TEA–TEB droplets at NPT conditions, implying the intrusion of oxygen to the condensed phase with the formation of (C_2_H_5_)_2_Al–O–O–(C_2_H_5_) and (C_2_H_5_)_2_B–O–O–(C_2_H_5_) molecules, seems plausible due to the indisputable experimental fact of the self-ignition of liquid TEA, TEB, or TEA–TEB mixture upon contact with air. The model shows that the rate of decomposition of these molecules with the formation of the active ethyl radical depends on the size of the liquid droplets, as well as the local instantaneous oxygen concentration and temperature in the environment, implying that the ignition delay is shorter for the higher injection pressure and hotter air. The results of preliminary experiments on the self-ignition of pulsed and continuous TEA–TEB sprays in air at NPT conditions have confirmed these implications and provided the data for estimating the activation energy of the rate-limiting reaction, which appeared to be close to 2 kcal/mol.

It must be noticed that the existence of reactions (5) and (6) and their supposed role in the TEA self-ignition process still remain questionable. For example, in [4], on the basis of molecular dynamics simulation of the reaction of TEA (in the condensed phase) with gaseous oxygen at a temperature above 2000 K, it is assumed that the reaction begins with the rapid removal of a hydrogen atom by an oxygen molecule, with the formation of the HO_2_ radical in the gas phase, and subsequently H_2_ molecules (the latter is associated with air humidity), i.e., the rapid occurrence of reactions in the gas phase is associated with the high reactivity of hydrogen. For the kinetic parameters A and ε in the analogue of formula (7), the values $9.67\cdot 10^{9}$ s^−1^ and 0.3 kcal/mol, respectively, were obtained. Another example is Ref. [19], where the priority is given to the channel of TEA decomposition through the breaking of the bond between oxygen atoms in the (C_2_H_5_)_2_Al–O–O–(C_2_H_5_) molecule with the formation of the C_2_H_5_O radical. In this case, the total process of oxygen intrusion and (C_2_H_5_)_2_Al–O–O–(C_2_H_5_) molecule decomposition is exothermic and proceeds with the release of 15.4 kcal/mol, whereas the same process proceeding through the C_2_H_5_ radical is claimed to be endothermic. The question of which mechanism is actually implemented can only be answered after systematic experiments, which are planned for the future. Note that the model proposed herein, even if it is not implemented in relation to the self-ignition of TEA, TEB, and TEA–TEB mixtures, can be applied to other reactions, the rate constant of which depends on temperature according to the Arrhenius law.

## 5. Conclusions

A novel scheme of the heterogeneous interaction of gaseous oxygen with liquid TEA–TEB droplets accompanied by the release of light hydrocarbon radicals C_2_H_5_ and/or C_2_H_5_O into the gas phase was used for calculating the self-ignition of a spatially homogeneous mixture of TEA–TEB droplets in ambient air at normal pressure and temperature conditions. Calculations were performed with the variation of the activation energy of the rate-limiting reaction intended for comparison with experiments. Calculations showed that the self-ignition delay of a stoichiometric C_2_H_5_O–air mixture was much longer than that of a stoichiometric C_2_H_5_–air mixture, indicating the much greater reactivity of the C_2_H_5_ radical compared to the C_2_H_5_O radical in air. The proposed kinetic model with the formation of (C_2_H_5_)_2_Al–O–O–(C_2_H_5_) or (C_2_H_5_)_2_B–O–O–(C_2_H_5_) molecules seems plausible due to the indisputable experimental fact of the self-ignition of liquid TEA, TEB, and TEA–TEB mixtures upon contact with air. Experiments on the self-ignition of pulsed and continuous TEA–TEB mixture sprays in air at normal pressure and temperature conditions provided the data for estimating the activation energy of the rate-limiting reaction, which appeared to be close to 2 kcal/mol. The ignition delay was shown to decrease with the decrease in the droplet size, both in the model and in the experiment.

## Figures and Tables

**Figure 1 micromachines-13-02033-f001:**
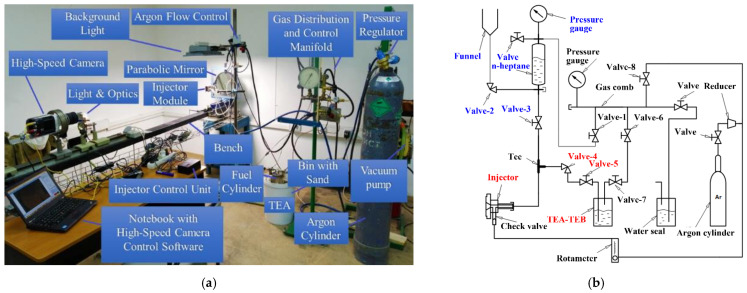
(**a**) Experimental setup and (**b**) the hydraulic scheme of fuel supply.

**Figure 2 micromachines-13-02033-f002:**
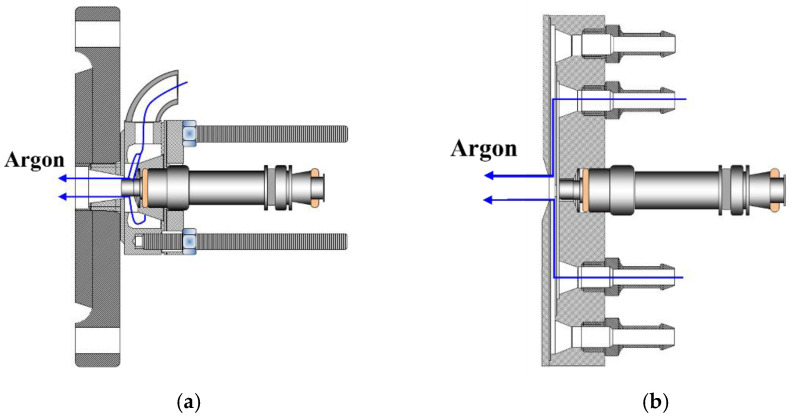
Two options of injector housing: (**a**) recessed injector and (**b**) flat injector.

**Figure 3 micromachines-13-02033-f003:**
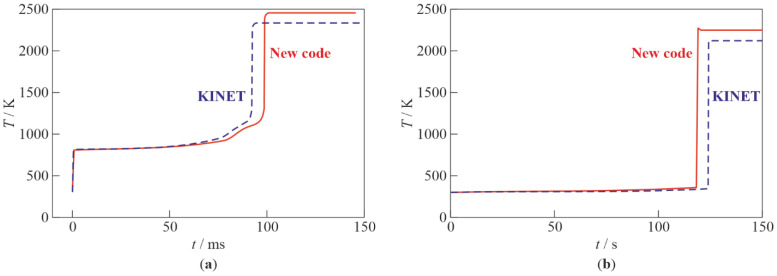
Calculated time histories of temperature during self-ignition of C_2_H_5_–air (**a**) and C_2_H_5_O–air (**b**) mixtures of stoichiometric composition at NPT conditions using two codes: new code and KINET [31].

**Figure 4 micromachines-13-02033-f004:**
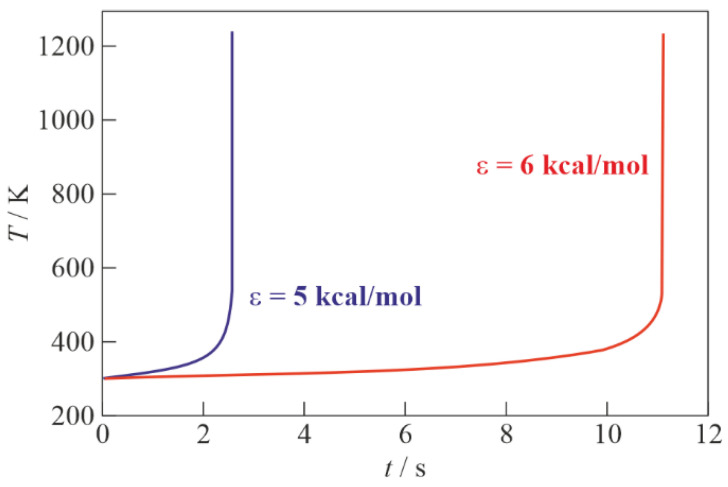
Calculated time histories of temperature during self-ignition of stoichiometric C_2_H_5_–air mixture with the heterogeneous reaction at NPT conditions and ε = 5 and 6 kcal/mol.

**Figure 5 micromachines-13-02033-f005:**
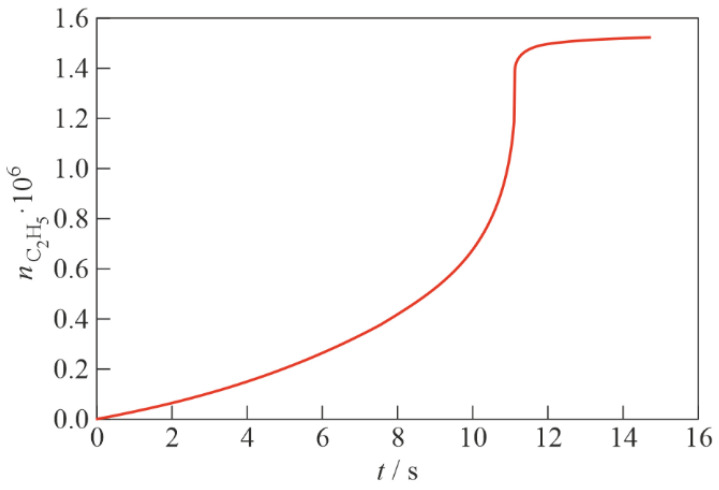
Calculated time history of the C_2_H_5_ radical concentration in a heterogeneous reaction at NPT conditions; ε = 6 kcal/mol.

**Figure 6 micromachines-13-02033-f006:**
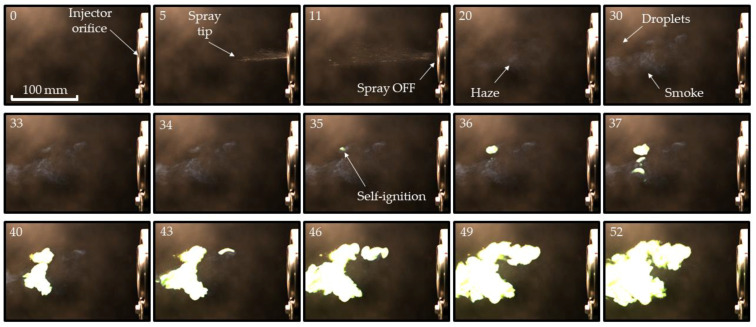
Sequence of video frames of pulsed TEA–TEB spray self-ignition and combustion in ambient air at NPT conditions: recessed injector. Frame numbers correspond to time in milliseconds from the start of injection. Injection overpressure is 3 atm. Frame size is 672 × 456 pixels (213 × 145 mm^2^), frame rate 1000 fps, shutter speed 400 μs.

**Figure 7 micromachines-13-02033-f007:**
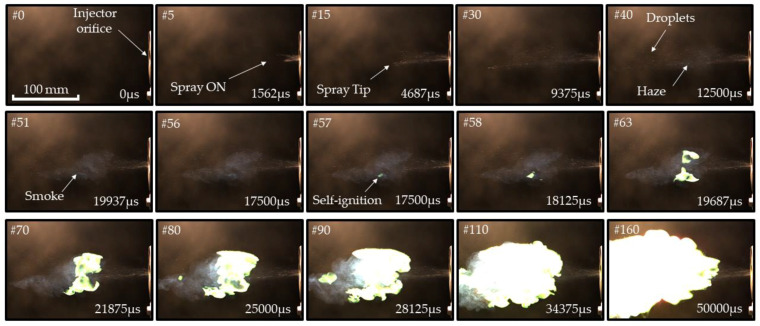
Sequence of video frames of continuous TEA–TEB spray self-ignition and combustion in ambient air at NPT conditions: recessed injector. Injection overpressure is 3 atm. Frame size is 672 × 456 pixels (213 × 145 mm^2^), frame rate 3200 fps, shutter speed 300 μs, injection duration 500 ms.

**Figure 8 micromachines-13-02033-f008:**
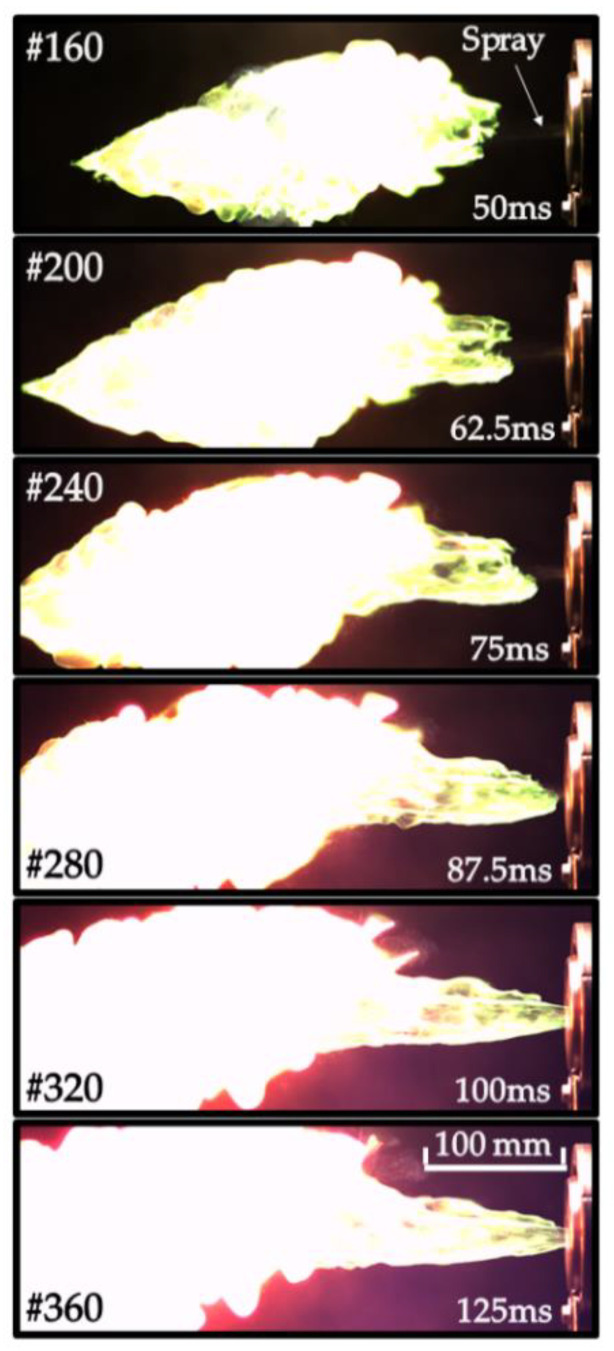
Sequence of video frames of continuous TEA–TEB spray combustion in ambient air at NPT conditions: recessed injector, the same test fire as in Figure 7. Injection overpressure is 3 atm. Frame size is 1280 × 456 pixels (406 × 145 mm^2^), frame rate 3200 fps, shutter speed 300 μs, injection duration 500 ms.

**Figure 9 micromachines-13-02033-f009:**
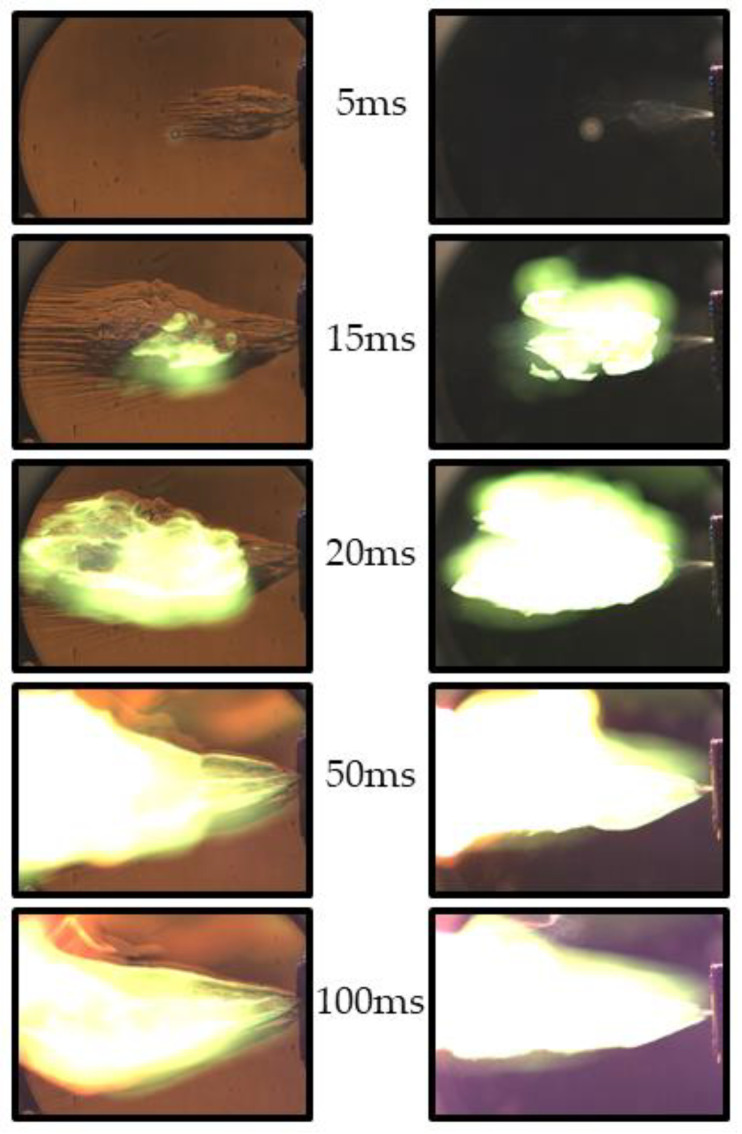
Sequence of schlieren ((**left**) column) and self-luminosity ((**right**) column) video frames of continuous TEA–TEB spray self-ignition and combustion in ambient air at NPT conditions: flat injector. Injection overpressure is 6 atm. Frame size is 590 × 392 pixels (192 × 127 mm^2^), frame rate 12,000 fps, shutter speed 81.5 μs, injection duration 100 ms.

**Figure 10 micromachines-13-02033-f010:**
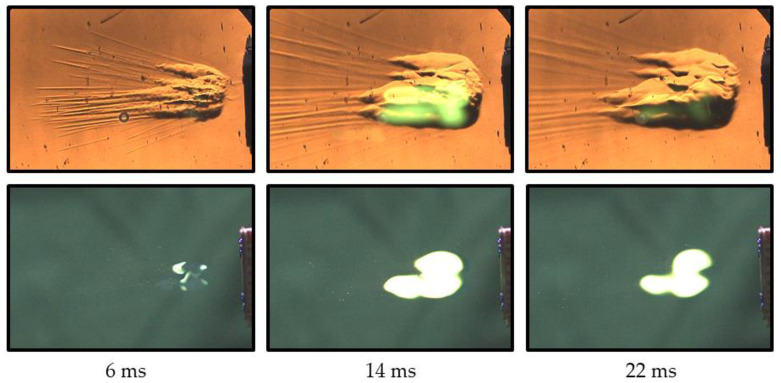
Sequence of schlieren ((**upper**) raw) and self-luminosity ((**lower**) raw) video frames of self-ignition and combustion of the pulsed TEA–TEB spray in ambient air at NPT conditions: flat injector. Injection overpressure is 6 atm. Frame size is 480 × 320 pixels (156 × 104 mm^2^), frame rate 5000 fps, shutter speed 190 μs, injection duration 0.7 ms.

**Table 1 micromachines-13-02033-t001:** Estimated values of the main variables.

Code	$\tau_{i},\;s$	$T_{e},\;K$	${\left(X_{C_{2}H_{5}}\right)}_{e}$	${\left(X_{O_{2}}\right)}_{e}$	${\left(X_{H_{2}O}\right)}_{e}$	${\left(X_{CH_{4}}\right)}_{max}$
New code	0.10	2455	0	0.007	0.14	0.0053
KINET	0.09	2336	0	0.010	0.13	0.0057

**Table 2 micromachines-13-02033-t002:** Self-ignition delays for option *i*.

ε, kcal/mol	$\tau_{i},\;s$
2	0.045
4	0.62
5	2.6
6	11.1

**Table 3 micromachines-13-02033-t003:** Measured self-ignition delays in the test fire with 4 successive injection pulses with a pulse duration of 10 ms and time interval between pulses 1000 ms; injection overpressure is 3 atm.

Test No.	1	2	3	4	Mean	RMS
$\tau_{i}$, ms	35	32	28	28	31	3

**Table 4 micromachines-13-02033-t004:** Measured self-ignition delays in the test fire with 6 successive injection pulses with a pulse duration of 5 ms and time interval between pulses 400 ms; injection overpressure is 3 atm.

Test No.	1	2	3	4	5	6	Mean	RMS
$\tau_{i}$, ms	24	20	19	19	17	17	19	2

**Table 5 micromachines-13-02033-t005:** Measured self-ignition delays in the test fire with 15 successive injection pulses with a pulse duration of 1 ms and time interval between pulses 100 ms; injection overpressure is 6 atm.

Test No.	1	2	3	4	5	6	7	8	9	10	11	12	13	14	15	Mean	RMS
$\tau_{i}$, ms	5	5.6	5.8	5.8	5.4	5.6	5.8	5.6	5.4	5.6	5.6	6	6.4	6	6.8	5.8	0.4

**Table 6 micromachines-13-02033-t006:** Measured self-ignition delays in the test fire with 15 successive injection pulses with a pulse duration of 2 ms and time interval between pulses 100 ms; injection overpressure is 6 atm.

Test No.	1	2	3	4	5	6	7	8	9	10	11	12	13	14	15	Mean	RMS
$\tau_{i}$, ms	5	5	4.8	5.2	5.4	5.2	5.2	5.8	5.8	6	5.4	6.2	6.2	6.2	6.2	5.6	0.5

## Data Availability

Not applicable.

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
