# Peer review of "Kinetic Model and Experiment for Self-Ignition of Triethylaluminum and Triethylborane Droplets in Air"

_micromachines, 2022, doi:10.3390/mi13112033_

Round 1

Reviewer 1 Report

This is a very exciting research exploration on the Triethylaluminum Al(C2H5)3, TEA, and triethylborane, B(C2H5)3, TEB, for the propellants of rockets. The paper presents the chemistry of the above with oxygen. I would like to recommend it to be published soon.

Author Response

This is a very exciting research exploration on the Triethylaluminum Al(C2H5)3, TEA, and triethylborane, B(C2H5)3, TEB, for the propellants of rockets. The paper presents the chemistry of the above with oxygen. I would like to recommend it to be published soon.

Dear reviewer,

Thank you very much for encouraging and stimulating comments.

With kind regards,

Authors.

Reviewer 2 Report

Comments

This paper conducts research on the self-ignition of triethylalu-

minum and triethylborane droplets in air. Nonetheless, there are still some problems in this paper that need to be improved.

1. The introductory section does not have enough current research status. The significance of conducting research needs to be highlighted.

2. The abstract and conclusion sections should be more coherent and clearly describe the results achieved.

 It is not recommended that this article be published in the Micromachines unless the above questions are well revised.

Author Response

We are grateful to the reviewer for valuable comments. We made our best to follow these comments. All changes in the revised manuscript are marked in green.

This paper conducts research on the self-ignition of triethylalu-minum and triethylborane droplets in air. Nonetheless, there are still some problems in this paper that need to be improved.

  1. The introductory section does not have enough current research status. The significance of conducting research needs to be highlighted.

To follow this comment, we have replaced reference 15 by a recent book (2022) and added a new sentence (and a new reference 16) to the Introduction section regarding the TEA significance for the modern industry:

According to [16], “Triethylaluminum market valued at 225.1 million USD in 2020 is expected to reach 255.2 million USD by the end of 2026, growing at a Compound Annual Growth Rate of 1.8% during 2021-2026.”

In addition, we have slightly rearranged the text in the Introduction section. We have moved some sentences above the last paragraph of the Introduction and highlighted the objectives of this work in a separate (last) paragraph.

In view of adding one new reference, we have renumerated all succeeding references in the manuscript.

  1. The abstract and conclusion sections should be more coherent and clearly describe the results achieved.

Following this comment, we have refined one sentence in the abstract

“The ignition delay is shown to decrease with the decrease in the droplet size,”

added a sentence to the Conclusion section:

The ignition delay was shown to decrease with the decrease in the droplet size both in the model and in the experiment.

 It is not recommended that this article be published in the Micromachines unless the above questions are well revised.

Reviewer 3 Report

The manuscript presented a novel method to calculate the  self-ignition of a spatially homogeneous mixture of TEA/TEB droplets under NPT condition.  The overall quality of this paper is high and I would have a few questions and comments regarding the content:

1. (page 2) Can the author explain why the QA values can be so different from each other? "The value of ?? is about 2444 kcal [2], 2293 kcal [17], 59 and 1955 kcal [18]." Any more reported QB values other than 2100 kcal[19]?

2. (page10) "Injection overpressure is 3 am. " should be 3atm

3. (page 4) "In the preliminary experiments, the 13%TEA-87%TEB mixture was used." How the TEA/TEB ratio was determined? If the  TEA/TEB ratio was lower or higher than the current value, how does is affect the experiments?

4. (page 7) "Figure 3 shows two options used for fastening the injector in the housing", "The first  series of experiments was performed using the recessed injector of Figure 3a and direct video registration of spray self-luminosity" and " the second series of experiments was performed using the flat injector of Figure 3b. " It's better to explicitly mention what are the first and second series of experience, or refer to the corresponding section number in the paper. 

5. (page 5) "two Options: (i) for a C2H5–air mixture and (ii) for a C2H5O–air mixture." were considered in experiments. Does it mean the situation of "C2H5/C2H5O coexistence-air mixture" is rare?

6. Could the author combine Figure 1 and Figure 2 into Figure 1 (a) and (b)?

7. In Figure 4 and Figure (5), "t/s" should be "t(s)", "t/ms" should be "t(ms)"   and "T/K" should be "T(K)". 

Author Response

Authors’ Response to Reviewer #3

We are grateful to the reviewer for valuable comments. We made our best to follow these comments. All changes in the revised manuscript are marked in gray.

The manuscript presented a novel method to calculate the self-ignition of a spatially homogeneous mixture of TEA/TEB droplets under NPT condition.  The overall quality of this paper is high and I would have a few questions and comments regarding the content:

  1. (page 2) Can the author explain why the QA values can be so different from each other? "The value of ?? is about 2444 kcal [2], 2293 kcal [17], and 1955 kcal [18]." Any more reported QB values other than 2100 kcal[19]?

Unfortunately, it is difficult to unambiguously explain the discrepancy between the values of QA for TEA reported by different researchers. Each method of QA estimation exhibits its own pros and cons. As for the value of QB, we could not find other more or less reliable values in the literature. Actually, the model we developed relies on the values of QA or Qb in general. However, the estimated value of the activation energy can certainly depend on the exact values of QA and QB according to the rule of Eq. (9).

  1. (page10) "Injection overpressure is 3 am. " should be 3atm

Done

  1. (page 4) "In the preliminary experiments, the 13%TEA-87%TEB mixture was used." How the TEA/TEB ratio was determined? If the  TEA/TEB ratio was lower or higher than the current value, how does is affect the experiments?

The standard TEA/TAB composition was provided by the production company. We did not change it in our experiments. We have mentioned it in the text as

“In the preliminary experiments, the standard 13%TEA-87%TEB mixture provided by the production company was used.”

  1. (page 7) "Figure 3 shows two options used for fastening the injector in the housing", "The first  series of experiments was performed using the recessed injector of Figure 3a and direct video registration of spray self-luminosity" and " the second seriesof experiments was performed using the flat injector of Figure 3b. " It's better to explicitly mention what are the first and second series of experience, or refer to the corresponding section number in the paper. 

We have referred explicitly here to the corresponding Figures in Section 3.3:

The first series of experiments was performed using the recessed injector of Figure 3a and direct video registration of spray self-luminosity during ignition and combustion in ambient air (see Figures 6 to 8 in Section 3.3). As the dark zone between the flame and the nozzle mouth could be quite short, the second series of experiments was performed using the flat injector of Figure 3b (see Figures 9 and 10 in Section 3.3).

  1. (page 5) "two Options: (i) for a C2H5–air mixture and (ii) for a C2H5O–air mixture." were considered in experiments. Does it mean the situation of "C2H5/C2H5O coexistence-air mixture" is rare?

To follow this comment, we have slightly reformulated the last sentences in Section 3.1:

“Thus, calculations show that the self-ignition delay of a stoichiometric C2H5O–air mixture is much longer than that of a stoichiometric C2H5–air mixture at NPT conditions. This indicates a much greater reactivity of the C2H5 radical in air compared to the C2H5O radical. In view of it the option with C2H5O radical can be omitted.”

  1. Could the author combine Figure 1 and Figure 2 into Figure 1 (a) and (b)?

Done. In view of it, we have renumbered all succeeding figures.

  1. In Figure 4 and Figure (5), "t/s" should be "t(s)", "t/ms" should be "t(ms)"   and "T/K" should be "T(K)". 

This is a matter of taste. We have recently published a paper in Micromachines with similar legends at the plots (see https://doi.org/10.3390/mi13091553)
